# Blind Deblurring of Remote-Sensing Single Images Based on Feature Alignment

**DOI:** 10.3390/s22207894

**Published:** 2022-10-17

**Authors:** Baoyu Zhu, Qunbo Lv, Yuanbo Yang, Xuefu Sui, Yu Zhang, Yinhui Tang, Zheng Tan

**Affiliations:** 1Aerospace Information Research Institute, Chinese Academy of Sciences, No.9 Dengzhuang South Road, Haidian District, Beijing 100094, China; 2School of Optoelectronics, University of Chinese Academy of Sciences, No.19(A) Yuquan Road, Shijingshan District, Beijing 100049, China; 3Department of Key Laboratory of Computational Optical Imagine Technology, CAS, No.9 Dengzhuang South Road, Haidian District, Beijing 100094, China

**Keywords:** remote sensing, image deblurring, generative adversarial networks, feature alignment, feature selection, deep learning

## Abstract

Motion blur recovery is a common method in the field of remote sensing image processing that can effectively improve the accuracy of detection and recognition. Among the existing motion blur recovery methods, the algorithms based on deep learning do not rely on a priori knowledge and, thus, have better generalizability. However, the existing deep learning algorithms usually suffer from feature misalignment, resulting in a high probability of missing details or errors in the recovered images. This paper proposes an end-to-end generative adversarial network (SDD-GAN) for single-image motion deblurring to address this problem and to optimize the recovery of blurred remote sensing images. Firstly, this paper applies a feature alignment module (FAFM) in the generator to learn the offset between feature maps to adjust the position of each sample in the convolution kernel and to align the feature maps according to the context; secondly, a feature importance selection module is introduced in the generator to adaptively filter the feature maps in the spatial and channel domains, preserving reliable details in the feature maps and improving the performance of the algorithm. In addition, this paper constructs a self-constructed remote sensing dataset (RSDATA) based on the mechanism of image blurring caused by the high-speed orbital motion of satellites. Comparative experiments are conducted on self-built remote sensing datasets and public datasets as well as on real remote sensing blurred images taken by an in-orbit satellite (CX-6(02)). The results show that the algorithm in this paper outperforms the comparison algorithm in terms of both quantitative evaluation and visual effects.

## 1. Introduction

Relying on the advantages of the spatial location of spacecraft platforms, such as satellites and space stations, the use of space optical cameras to observe the Earth has the advantages of a wide range of detection, fast data acquisition, a large amount of information, and less restrictions induced by geographical and natural conditions. It is an important way to obtain information for global change monitoring, regional surveillance, etc. It has been applied in many fields, such as in land and mineral resources management and monitoring, traffic and road network safety monitoring, early warning systems geological disasters, national defense system construction, and cosmic origin research [1]. However, as the demand for the geometric resolution of aerospace optical camera images continues to rise, capturing high-quality images has also encountered great challenges, as highlighted by the problem of avoiding image quality degradation due to image shift blur at high speeds and random vibrations [2].

To reduce the blurring of image shift caused by the high-speed space motion of platforms, such as satellites and space stations, they are usually compensated using a motion compensation mechanism. To ensure that the relative motion of the imaging sensor is less than one-half of an image element or less during the exposure time, this compensation method increases the development cost of the spacecraft. At decimeter resolution, motion compensation accuracy is required to a higher degree, and the corresponding motion compensation mechanism becomes complex [3]. Furthermore, spacecraft vibrations in space cannot be avoided, and micro-vibrations can cause complex imaging blurring when reaching decimeter resolution. High-precision vibration isolation techniques and devices [4] are very complex, costly, and take up substantial weight and space, reducing the cost-effectiveness ratio of the satellite. Therefore, the use of data processing to repair motion blur in remote sensing images can significantly reduce the requirements for satellite motion compensation mechanisms and vibration isolation mechanisms or may take on some of the functions of image shift compensation, improving the efficiency ratio of the spacecraft, which has important application value.

The common formula [5] for nonuniform fuzzy models is as follows:(1)B=S∗k(M)+N
where *B* is the blurred image, *S* is the corresponding clear image, *k*(*M*) denotes the unknown blur kernel, *M* denotes a sparse matrix with each row containing a local fuzzy kernel, ∗ is the discrete convolution, and *N* is the noise. Deblurring methods are generally divided into two categories: non-blind image deblurring and blind image deblurring. Non-blind deblurring algorithms assume that the blur kernel is known or pre-estimated, whereas blind deblurring algorithms estimate both the blur kernel and the clear image or output the clear image directly. Early conventional deblurring models used natural image priors to design the corresponding algorithms [6,7,8]; however, when dealing with images with null-variant properties, most a priori models do not capture the complex blurring variations in real images well. To ameliorate this problem, a number of deep learning-based deblurring algorithms have been proposed, with neural networks themselves offering unique advantages that are unmatched by traditional algorithms:They generate images directly from the network without estimating fuzzy kernels, avoiding the problem of error superposition during the estimation of fuzzy kernels;There is no need to consider whether the motion blur has a spatially varying property, extending the applicability of the algorithm.

Therefore, from the initial stages of the development of neural network algorithms, researchers have applied neural networks to the problem of processing image deblurring with good results [9].

The recent introduction of generative adversarial networks (GANs) [10] has greatly facilitated the development of deep learning, which has led to breakthroughs in several areas of image processing. In particular, in the field of image restoration, GANs often produce sharper and more plausible textures than feedforward encoders [11]. DeblurGAN [5] introduces GAN to the deblurring task, treating image deblurring as a special image-to-image conversion task, where the model can recover clear and visually appealing images of good quality from both synthetic and real blurred images. DeblurGAN-v2 [11] builds on the success of DeblurGAN by constructing a new cGAN framework. A feature pyramid network (FPN) with multiscale feature fusion was introduced in the generator, global (image) and local (patch) scales were introduced in the discriminator, and RaGAN-LS loss was proposed to improve the deblurring performance and inference efficiency.

However, due to the unlearnable nature of upsampling operations commonly used in FPN (e.g., nearest neighbor interpolation) and the reuse of downsampling and upsampling, there is a problem of feature misalignment between downsampling and upsampling. The unaligned features in turn adversely affect learning in subsequent layers, resulting in the poor recovery of the final blurred image, especially in terms of detail. One approach to solving feature alignment is to perform alignment by explicitly estimating the optical flow field between a reference frame and its neighboring frames [12]. A further approach is to implement implicit motion compensation through circular contrast [13] or deformable convolution [14]. Spatially varying blurred images presents a significant challenge to existing alignment algorithms. For stream-based methods, in particular, stream estimation and image-level warping strategies are very time-consuming and prone to artifacts [14]. In the presence of large image motion, it is difficult to compensate for motion explicitly or implicitly within a single resolution range.

This paper aims to provide another substantial improvement to the cGAN-based motion deblurring task. To address the problem of feature misalignment, feature selection and feature alignment modules are introduced to improve the structure of the generator network, called SDD-GAN, which improves the deblurring performance of the model.

The innovations in this paper are summarized below.

Feature importance selection module

Remote sensing images are rich in content, and, to obtain useful features, this paper proposes FISM, which constructs feature attention modules in the spatial and channel domains, explicitly models the interdependence between features, emphasizes lower-level features with rich spatial details, obtains reliable detail information in remote sensing images, achieving the purpose of feature importance selection, and helps to recover details in remote sensing blurred images.

2.Feature alignment fusion module

Multiscale feature extraction and fusion can effectively deal with the complex and variable motion blur of remote sensing images, but multiscale fusion can result in feature misalignment. This paper proposes FAFM, which learns to align and fuse the feature map of the upsampled channel with the corresponding reference feature map of another channel by adjusting each sampling position in the convolution kernel using the learned offsets. FAFM can better aggregate feature information from different spatiotemporal locations for blurred image recovery.

3.Color feature loss function

This paper introduces a color loss function to minimize the error between the clear image and the reconstructed deblurred image, thereby forcing the generator to produce an image with the same color distribution in detail as the clear image. The perceptual loss function was also optimized, and the perceptual model VGG19 was pre-trained on ImageNet [15] followed by further training on the RSDATA dataset and the dichotomous GoPro dataset so that the VGG19 network could better obtain the desired image features.

4.RSDATA dataset

In this paper, according to the imaging mechanism of remote sensing satellites, the types of blurring that may occur and the offset range of pixels in the blurred images are analyzed by detailed calculations. Accordingly, a self-built remote sensing dataset RSDATA is constructed, with the raw data coming from the publicly available datasets UCAS-AOD and DOTA for remote sensing. On the test set, the algorithm in this paper outperformed all comparative algorithms in terms of quantitative evaluation and visual effects, which demonstrates the important role and promise of feature alignment in the field of remote sensing image deblurring.

## 2. Related Work

### 2.1. Image Deblurring-Based Deep Learning

Deep learning methods have achieved great success in several image processing tasks including image deblurring. Sun et al. [16] were the first to introduce convolutional neural networks (CNNs) into the field of image deblurring, significantly changing the poor results of fully connected network-based image deblurring algorithms. DeepDeblur [17] proposed a multiscale neural network to iteratively restore clear images, introducing the idea of generative adversarial networks [10]. DeblurGAN [5] proposed an image deblurring algorithm based on generative adversarial networks using gradient penalty and perceptual loss to achieve good recovery of blurred images using the GAN’s better texture generation in the generated images. DeblurGan-V2 [11] performed an optimization of the initial DeblurGAN by introducing a lightweight backbone network to improve the speed of the model and introducing a feature pyramid network [18] into the image deblurring model, which can work flexibly with various backbone networks to achieve a balance between performance and efficiency. SAPHNet [19] proposed an efficient deblurring design built on a new convolutional module that uses global attention and adaptive local filters to learn the transformation of features. MAXIM [20] proposed a multi-axis MLP-based architecture with global/local perceptual fields in each module to improve the learning capability of the model. Ref-MFFDN [21] proposed a new reference-based multilevel feature fusion deblurring network for remote sensing image deblurring in which multi-level features are fused in the same location of clear reference images at different moments in time to extract textures and help recover blurred images. However, these deblurring algorithms do not deal well with the problem of feature misalignment in the process of multi-layer feature fusion.

### 2.2. Feature Alignment

As the convolutional neural network deepens, the feature map gradually shrinks, causing severe loss of detailed information and leading to severe misalignment during feature fusion. Dai et al. [22] first proposed deformable convolution to learn additional offsets to allow the network to obtain information far from its prescribed local neighborhood, thus improving the ability to convolve. TDAN [14] used deformable convolution to align input frames without explicit motion estimation or image warping. EDVR [23] proposed a deformable convolution-based PCD module for feature alignment, which effectively circumvents the need to compute/estimate image optical flow explicitly or implicitly in traditional alignment methods. FaPN [24] proposed a feature alignment module for dense prediction tasks, which learns the transform offset of pixels, aligns upsampled high-level features according to context, and integrates them in a top-down pyramidal architecture. AlignSeg [25] proposed a two-branch bottom-up network that uses two types of alignment modules to solve the feature misalignment problem prior to feature aggregation, including an aligned feature aggregation (AlignFA) module and an aligned context modeling (AlignCM) module. AlignFA uses a learnable interpolation method to learn the transform offset of pixels, and AlignCM enables each pixel to select its contextual information adaptively so that the contextual information can be aligned and fused, which attempts to directly align features of different proportions.

### 2.3. Feature Selection

Convolutional neural networks are built on convolutional operations that fuse spatial and channel information within the local receptive field to extract information features. Recent research has shown that the performance of the network can be improved by explicitly embedding learning mechanisms that help capture spatial correlations and improve the representational power of the network without additional supervision. Attention mechanisms have been used with great success in many visual tasks, such as channel attention, spatial attention, temporal attention, and branching attention. In the visual system, attentional mechanisms can be considered as a dynamic selection process, achieved by adaptively weighting features according to the importance of the input.

SENet [26] proposed the channel-attention network, which can adaptively predict potential key features and recalibrate channel feature importance by explicitly modeling the interdependencies between channels. GSoP-Net [27] improved the ability to collect global information through SEBlocks by using global second-order pooling (GSoP), GsoPBlock, but it is computationally intensive. ECANet [28] replaced the fully connected layer in SENet with a one-dimensional convolution. GCT [29] proposed a generic and lightweight change module that combines normalization methods and attention mechanisms to analyze the interrelationships between channels. FcaNet [30] proposed an approach with multispectral channel attention. FaPN [24] proposed a feature selection module (FSM) to explicitly model the importance of feature maps, introducing additional jump connections between input and scaled feature maps for necking (i.e., top-down paths) to enhance multiscale feature aggregation.

## 3. Network Structure of SDD-GAN

As shown in Figure 1, the SDD-GAN in this paper consisted of two main parts: a generator and a discriminator. The main role of the generator is to generate clear images. The main role of the discriminator is to judge the authenticity of the generated images.

### 3.1. Generator Network Structure of SDD-GAN

An overview of the architecture of the SDD-GAN generator is shown in Figure 2. SDD-GAN restores a single blurred image to a clear image by training the generator. The generator is the core part of the entire generative adversarial network, and the performance of this part determines the quality of the final recovery of the blurred images.

DeblurGAN-v2 [11] proposed a lightweight alternative to multiscale functionality by introducing the feature pyramid network (FPN) [18] structure into the generator. In this paper, we used a feature pyramid similar to DeblurGAN-v2 as the base structure for feature extraction to build a generator for SDD-GAN, including two channels: a bottom-up channel that downsamples the spatial resolution, and a top-down channel that reconstructs a higher-spatial-resolution image, which can generate multiple feature layers, encode different semantics, contain better-quality feature information, and add lateral connections between the bottom-up and top-down paths to complement the high-resolution detail.

The nearest neighbor upsampling in the top-down channel of DeblurGAN-v2 is modified to a null convolution operation, and a feature importance selection module and a feature alignment fusion module are added to enhance the capability of multiscale feature extraction and aggregation.

Separate feature extraction for images of different spatial resolutions followed by upsampling of high-level feature information, feature selection, and feature alignment followed by feature fusion can well extract the detailed information and global information needed to restore the image, so as to generate a feature map with good semantic information in each spatial scale of the image, and reduce the error or loss of details during feature fusion. Five final feature maps at different scales are selected as stage outputs in the top-down channel. After these features are selected by the feature importance selection module, *W* and *H* are upsampled to the same size and stitched into a tensor that contains different levels of semantic information. Two additional upsampling and convolution layers are added at the end of the network to recover the original image size and reduce artifacts. Jump connections are introduced from input to output to improve network performance and convergence, with the input image being normalized to [−1, 1] during training, and the output is kept in the same range using the tanh activation layer.

### 3.2. Feature Alignment Fusion Module

FaPN [24] proposed a feature alignment fusion module for dense prediction tasks, which learns the transform offset of pixels by deformable convolution. Inspired by it, for the problem of inaccurate correspondence between FPN downsampled and upsampled features, this paper proposes a feature alignment module based on defuzzification model properties and an improved generator structure.

This module learns to align the feature map of the upsampled channel with the corresponding reference feature map of another channel by adjusting each sampled position in the convolution kernel with the learned offset. This paper further proposes a feature selection module that adaptively emphasizes feature maps that contain rich detail. The two modules are then integrated into a top-down pyramidal architecture that progressively aligns features from the coarsest resolution (top) to the finest resolution (bottom), as shown in Figure 3.

The level I network channel of the bottom-up reduced feature map is defined as FiA, and the level II network channel of the top-down enlarged feature map is defined as FiB. Due to the downsampling operation, there is a predictable spatial misalignment between the upsampled feature map F˙iB and the corresponding bottom-up feature map Fi−1A. Therefore, feature fusion by direct correspondence summation of elements or channel cascading compromises the accuracy of feature map detail. Prior to feature fusion, aligning the FiB with its reference Fi−1A is an essential part of the process. The FiB is adjusted according to the spatial location information provided by the Fi−1A. Specifically, the feature map FiA of the level I network has a span of 2*^i^* pixels relative to the input image.
(2)FiA∈ℝH2i×W2i
where *H* × *W* is the size of the input image. For brevity, we denote H/2i,W/2i as (hi,wi).

The output of the fusion of the first feature map $F_{i+1}^{B}$ in the top-down path with the $F_{i}^{A}$ of the level I network is defined as $F_{i}^{B}$. Specifically, the feature of $F_{i+1}^{B}$ after deconvolution is defined as ${\dot{F}}_{i+1}^{B}$, ${\dot{F}}_{i+1}^{B}$, and $F_{i}^{A}$ after the first FISM layer to obtain ${\ddot{F}}_{i+1}^{B}$ and ${\ddot{F}}_{i}^{A}$. The ${\ddot{F}}_{i+1}^{B}$ feature is aligned to the $F_{i}^{A}$ feature by calculating the offset between ${\ddot{F}}_{i+1}^{B}$ and $F_{i}^{A}$, and the feature map is defined as ${\dddot{F}}_{i+1}^{B}$. ${\dddot{F}}_{i+1}^{B}$ and $F_{i}^{A}$ go through a second FISM layer for feature selection to obtain the feature maps ${\overset{ˇ}{F}}_{i+1}^{B}$ and ${\overset{ˇ}{F}}_{i}^{A}$. Finally, the feature maps ${\overset{ˇ}{F}}_{i}^{A}$ and ${\overset{ˇ}{F}}_{i+1}^{B}$ are fused to obtain $F_{i}^{B}$.

In the feature alignment module, spatial location information is represented by 2D feature maps, where each offset value can be considered as the offset distance in 2D space between each point in ${\dot{F}}_{i+1}^{B}$ and the corresponding point in $F_{i}^{A}$; the calculation flow for feature alignment is illustrated in Figure 4.

The feature alignment can be expressed mathematically as follows:(3)$${\dddot{F}}_{i+1}^{B}=f_{a}\left({\ddot{F}}_{i}^{B},\Delta_{i}\right)$$
(4)$$\Delta_{i}=f_{o}\left(\left[\lambda_{B}{\ddot{F}}_{i+1}^{B},\lambda_{A}{\ddot{F}}_{i}^{A}\right]\right)$$

$\left[\lambda_{B}{\ddot{F}}_{i+1}^{B},\lambda_{A}{\ddot{F}}_{i}^{A}\right]$ is a cascade of ${\ddot{F}}_{i+1}^{B}$ and ${\ddot{F}}_{i}^{A}$ ($\lambda_{A}$, $\lambda_{B}$ are hyperparameters), which provides the spatial difference between the corresponding top-down feature $F_{i+1}^{B}$ and the corresponding bottom-up feature $F_{i}^{A}$. $f_{a}\left(\;,\right)$ denotes the learning offset $\Delta_{i}$ according to the spatial difference between the two feature maps, the two feature maps are concatenated to obtain the offset of the *XY*-axis through convolution, and the parameters are updated through backpropagation during training. $f_{o}\left(\;,\right)$ denotes the function that aligns the feature map in the *XY*-axis direction according to the learned offset $\Delta_{i}$. In this paper, $f_{a}\left(\;,\right)$ and $f_{o}\left(\;,\right)$ are implemented using deformable convolution [31].

### 3.3. Feature Importance Selection Module

In this paper, the FISM module is proposed to improve the generator model, with the data flow shown in Figure 5. The feature attention module is constructed in the spatial and channel domains to explicitly model the interdependencies of features, emphasizing lower-level features with rich spatial detail for feature importance selection. As shown in Figure 2 and Figure 4, the FISM module is added to the four jump connections, as well as the feature alignment fusion module in this paper.

First, the information $z_{w},z_{c},\;\mathrm{and}\;z_{h}$ of each input feature Pi mapping in the spatial and channel domains is extracted through a pooling operation, and then feature importance modeling $f_{w},f_{c},\;\mathrm{and}\;f_{h}$ in each of the three domains (i.e., two 1 × 1 Conv layers with a ReLU activation function in between) is performed, using this information to learn to model the importance of each feature mapping in the three domains to ensure that the network can improve its sensitivity to the informative features and output the corresponding three importance vectors $S_{w},S_{c}$, and $S_{h}$, followed by a sigmoid activation function. Next, the original input feature map is scaled using importance vectors, and the scaled feature map is then added to the original feature map. Finally, a feature selection layer $F_{s}\left(\;.\;\right)$ (using 1 × 1 Conv to weight each layer of features) is introduced over the rescaled feature map to selectively retain important feature maps and suppress useless feature maps to reduce the amount of information.

The FISM process can be simplified using the following formula:(5)$$P_{i}^{\prime }=F_{s}(P_{i}+S_{w}*S_{c}*S_{h}*P_{i})$$
(6)$$\begin{array}{l} S_{w}=f_{w}(z_{w})\\ S_{c}=f_{c}(z_{c})\\ S_{h}=f_{h}(z_{h}) \end{array}$$
where $z^{a}=\left[z_{1},z_{2},\;\ldots ,\;\;z_{D}\right]\;,\;a\in \left(W,C,H\right)$, and $z^{a}$ is calculated as follows:(7)$$z_{d}^{a}=\frac{1}{H_{i}^{a}\times W_{i}^{a}}\sum_{h^{a}=1}^{H_{i}^{a}}\sum_{w^{a}=1}^{W_{i}^{a}}c_{d}^{a}(h^{a},w^{a})$$

### 3.4. Loss Function

To train the algorithm in this paper, it is necessary to compare the image in the training phase, the reconstructed image, and the original image under certain metrics. We designed a loss function *L* consisting of four components and minimized it during the training of the model, expressed as
(8)$$L=w_{g}L_{g}+w_{p}L_{p}+w_{x}L_{x}+w_{c}L_{c}$$

This paper continues the partial loss function of DeblurGAN-v2, choosing the mean square error (MSE) loss as $L_{p}$, and using the RaGAN-LS [11] loss as the global and local discriminator loss function $L_{g}$, to make the training process smooth and efficient. *D* denotes discriminator, *G* denotes generator, $I_{X}$ denotes the blurred image domain, and $I_{Y}$ denotes the clear image domain, where *x* is the blurred image and *y* is the corresponding clear image:(9)$$\begin{array}{r} L_{g}=E_{x\sim I_{X\;}(x)}\left[{\left(D(x)-E_{y\sim I_{Y}(y)}D(G(y))-1\right)}^{2}\right]\\ +E_{y\sim I_{Y}(y)}\left[{\left(D(G(y))-E_{x\sim I_{X\;}(x)}D(x)+1\right)}^{2}\right] \end{array}$$

Perceptual loss [32] is based on the *L2* loss of the CNN feature maps of the generated and target images, which measures the high-level perceptual gap and the semantic gap between the two images and is widely used in image transformation tasks. The perceptual loss used in this paper is the difference between a clear image and a recovered blurred image in the conv3.3 feature map of VGG-19.
(10)$$L_{X}=\frac{1}{W_{i,j}H_{i,j}}\sum_{w=1}^{W_{i,j}}\sum_{h=1}^{H_{i,j}}{\left(\phi_{i,j}{\left(Y\right)}_{w,h}-\phi_{i,j}{\left(G\left(X\right)\right)}_{w,h}\right)}^{2}$$
where *i*, *j* denotes the feature map ordinal number of the *i*-th max-pooling layer followed by the *j*-th convolution (after activation) within the VGG19 network, $\phi_{i,j}$ denotes the feature map, and $W_{i,j}$ and $H_{i,j}$ are the dimensions of this feature map.

This paper attempts to pre-train VGG19 on ImageNet, the RSDATA dataset, and the dichotomous GoPro dataset, which allows the VGG19 network to better obtain the desired image features while calculating perceptual losses more efficiently.

For the image transformation task, CycleGAN [33] attempts to add a loss between *G*(*Y*) and *Y*, called constant loss. DeepUEP [34] uses color loss to drive the color of the generated image to match the color of the corresponding clear image. On this basis, this paper introduces a color loss function to minimize the error between the clear image and the reconstructed deblurred image, as a way to force the generator to generate images with the same color distribution in detail as the clear image, with the color feature loss function defined as follows:(11)$$L_{c}^{}=E_{x\sim I_{x\;}(x),}{}_{y\sim I_{Y\;}(y),}\left[\sum_{p}∠({\left(G(x)\right)}_{p},{\left(y\right)}_{p})\right]+E_{y\sim I_{Y\;}(y)}\left[{\left\|G(y)-y\right\|}_{1}\right]$$

Treating the RGB color space of an image as a 3D vector space, where ∠(,) is the operator that calculates the angle between two images in vector space, ${\left(\;.\right)}_{p}$ denotes a pixel point of an image, *x* denotes a blurred image in domain *X*, and *y* denotes a clear image in domain *Y* corresponding to *X*. *G*(*x*) denotes the image after the blurred image has been deblurred, and *G*(*y* denotes the image after the clear image has been deblurred. This paper addresses the problem of detailed color distortion during image deblurring by optimizing the sum of errors per pixel point between the deblurred image and the paired clear image. This paper uses the L1 loss function to optimize the error of the clear image and *G*(*y*, which can limit the change interval of the generator in the update iteration and make the details of the generated image of the generator more stable and reliable, while having a certain data-widening ability, thus prompting the generator to better restore the color detail features of the image and improve the quality of the generated image.

### 3.5. Remote Sensing Image Datasets

Motion blur is a common type of image blur in the field of remote sensing. It mainly stems from the irregular movement of the imaging device during exposure. There are no publicly available datasets for remote sensing image deblurring tasks; hence, this paper analyzes scenes that may cause blurring based on the actual process of remote sensing satellite photography, along with detailed calculations of the types of blurring that may occur and the offset range of pixels in the blurred image. The calculation process is described below.

Assume that the rotation speed of the remote sensing satellite around the Earth is ${\vec{V}}_{s}$, the Earth’s rotation speed is ${\vec{V}}_{e}$, $\vec{\vartheta }$ is the correlation coefficient between the rotation of the Earth and the velocity of the satellite at the time of the satellite shot, the exposure time of the satellite photography is $T_{p}$, the compensation of the satellite motion is Ψ, the radius of the Earth is *R*, the distance of the satellite from the surface is *r*, the resolution of the satellite image is *Res*, and the error of the satellite motion on the photography image during the exposure time is ${\vec{L}}_{M}$, as shown in Equation (12).
(12)$${\vec{L}}_{M}=\frac{\left({\vec{V}}_{s}+\vec{\vartheta }\;{\vec{V}}_{e}\right)\times R\times T_{p}}{\left(R+r\right)\times Res\times \Psi }$$

Camera shake can affect satellite photography and blur the image [35]; in this paper, we only study the blurred phase shift caused by this error and define it as ${\vec{L}}_{D}$. The total error in the exposure time of the ground-based static target is calculated using Equation (13).
(13)$${\vec{L}}_{static}={\vec{L}}_{D}+{\vec{L}}_{M}.$$

Assuming that the velocity of the ground-based moving target is ${\vec{V}}_{obj}$ and that $\vec{\rho }$ is the correlation coefficient between the velocity of the moving target and the velocity of the satellite, the error caused by the moving target during the shot exposure time is ${\vec{L}}_{obj}$, as expressed in Equation (14):(14)$${\vec{L}}_{obj}=\vec{\rho }\frac{{\vec{V}}_{obj}\times T_{p}}{Res\times \Psi }$$

The blurred images generated by DeepDeblur [17] exhibit real and spatially varying blur caused by moving people and static backgrounds; by the same token, the error in remote sensing satellite photography can be simplified as
(15)$${\vec{L}}_{ALL}=\left\{\begin{array}{c} \;\;\;\;\;\;\;{\vec{L}}_{static}\;\;\;\;\;\;\;,\;\;\;\;background\\ {\vec{L}}_{static}+{\vec{L}}_{obj}\;,\;\;\;\;moving\;object\;and\;background\; \end{array}\right.$$

Let the upper left corner of the remote sensing image be the origin, the horizontal axis be the *X*-axis, the vertical axis be the *Y*-axis and the angle between the error vector and the *X*-axis be θ. The phase shift of pixels on the image can be expressed as
(16)$$\left\{\begin{array}{c} \Delta x={\vec{L}}_{ALL}\times cos\theta \\ \Delta y={\vec{L}}_{ALL}\times sin\theta \end{array}\right.$$

Remote sensing images taken by satellite have both null-variant and non-null-variant types of blur. This paper constructs a remote sensing deblurring task dataset (RSDATA) on this basis, with raw data collected from the remote sensing open datasets UCAS-AOD [36] and DOTA [37]. The UCAS-AOD dataset was collected from Google Earth and the images for the DOTA dataset were collected from Google Earth, GF-2, and JL-1 satellites provided by the China Resource Center Satellite Data and Applications. The dataset is rich in content and includes multiple regions around the world, including airports, vegetation, ports, and high-density urban areas, scaling all images to 0.5 M resolution based on the GSD originally provided with the dataset. The areas were further cropped to a size of 540 × 540 pixels, overlapping by 10 pixels. To simulate a more realistic and complex motion blur process, the random trajectory generation method proposed in DeblurGAN is used in this paper to generate motion blur kernels to generate blurred images; according to Equation (15), the image blur phase shift size varies randomly in the (5, 41) interval on which based 10% of the data are randomly selected to add local null blur, and the phase shift varies randomly in the (1, 5) interval. Eighty percent of the dataset was used as the training set and 20% was used as the test set. The training set contained 1330 pairs and the test set contained 340 pairs, with each pair consisting of a clear image and a corresponding blurred image. An example of our dataset is shown in Figure 6.

## 4. Results

To verify the generalizability of the training results, comparative experiments were not only conducted on the self-built remote sensing dataset RSDATA but also on the GoPro dataset [17] and RealBlur dataset [38].

To quantitatively assess the performance of different network models, recovered images were evaluated using the peak signal-to-noise ratio (PSNR) and structural similarity metric (SSIM), which are widely used metrics for image quality assessment.

This paper conducts an ablation study in Section 4.3 to verify the effectiveness of the feature alignment fusion module, explore the impact of adding a feature importance selection module with multiple domains on the algorithm and confirm the effectiveness of the color feature loss function.

### 4.1. Datasets

GoPro dataset: The GoPro dataset [17] is currently one of the most commonly used benchmark datasets in the field of image deblurring based on neural network algorithms. The dataset is derived from the results of frame decomposition of HD video images and is rich in content, covering almost all common image features. The entire dataset has a total of 3214 blurred/clear image pairs, each being 720 × 1280 pixels in size. The dataset itself is divided into three parts: Train, Valid, and Test, with a total of 2103 images in the Train group used for training;RealBlur dataset: RealBlur-R [38] is generated from raw camera images and contains 232 different low-light still scenes with a total of 4556 pairs of blurred and sharp images. The images in the dataset are blurred due to camera shake and captured in dimly lit environments (e.g., streets and indoor rooms at night) to cover the most common cases of motion blur, representing blurred images that are more in line with the real world. RealBlur-R randomly selected 182 scenes as the training set, consisting of 3758 image pairs, with the remaining 50 scenes as the test set, consisting of 980 image pairs;Real satellite images: In-orbit real-world data were obtained from the CX-6(02) optical remote sensing satellite, which was launched on 22 December 2016 with an imaging resolution of 2.8 m in a 700 km orbit. In addition to ground-based remote sensing imaging, experimental verification of image super-resolution processing [39] and image deblurring was carried out using CX-6(02) satellite image data. For the deblurring experiments, the satellite attitude was controlled to obtain image shift blurred image data of approximately 25 pixels, with no clear images for pairing, and this set of image data was used to verify the effectiveness of the deblurring algorithm on the ground;RSDATA: In this paper, a remote sensing deblurring mission dataset was constructed with raw data collected from the remote sensing open datasets UCAS-AOD [36] and DOTA [37], details of which were described in Section 3.5.

### 4.2. Comparative Results

To evaluate the performance of the algorithm in this paper, the method in this paper was compared with other SOTA deblurring methods on three datasets. All comparison procedures and test results were taken from the authors’ official websites.

RSDATA dataset: This paper quantified the results of DeepDeblur [17], DeblurGAN-v2 [11], DBGAN [40], and MAXIM [20] on a self-constructed RSDATA dataset.

The official websites of the comparison algorithms all provide the model weights of the algorithms on the GoPro dataset. Therefore, in this paper, we first chose to train on the GoPro dataset (trained on GoPro) and test on the RSDATA dataset. The average PSNR and average SSIM of different deblurring algorithms are shown in Table 1. The algorithm in this paper outperformed other deblurring methods in terms of average PSNR and average SSIM. The algorithm in this paper achieved a PSNR of 19.96 dB, which was 1.6 dB better than DeblurGAN-v2.

Figure 7 shows the statistical results of comparing the PSNR and SSIM of the algorithms on the RSDATA test set (trained on GoPro). As shown in Figure 7a, the PSNR metric of the recovered images of this paper’s algorithm was higher than that of the other algorithms at the maximum, minimum, and median values. Figure 7b shows that the images recovered by SDD-GAN not only had higher values than the other deblurring algorithms but also had the smallest range of values for the SSIM metric, demonstrating the robustness of SDD-GAN on the RSDATA test set.

A qualitative assessment of the results of the different algorithms is shown in Figure 8. The remote sensing images recovered by DeepDeblur and DeblurGAN-v2 were blurred and lacked texture. The DBGAN approach could recover clear images, but the recovered images also had ghosting, artifacts, and false textures. MAXIM generated images with a good visual appearance, but the results were still poor when recovering some fine textures (e.g., text on airport runways). In contrast, the method proposed in this paper could produce images with fine textures and realistic visual effects, whether for text, vegetation, or buildings.

Trained and tested on the RSDATA dataset, the algorithm in this paper achieved a PSNR of 39.88 dB and an SSIM of 0.92. Compared with training on GoPro, the test results get a substantial improvement, indicating that the non-remote sensing image dataset GoPro was relatively different from the remote sensing fuzzy dataset; thus, a new remote sensing fuzzy dataset is necessary.

GoPro dataset: The results of this paper were compared quantitatively with other methods on the GoPro dataset, including DeblurGAN [5], DeepDeblur [17], SRN [41], STFAN [42], DeblurGAN-v2 [11], and BlurredItV [43].

The average PSNR and average SSIM of different deblurring algorithms trained and tested on the GoPro dataset are shown in Table 2. The algorithm in this paper outperformed other defuzzification methods in terms of average PSNR and average SSIM. The algorithm in this paper achieved a PSNR of 30.95 dB, which was 1.4 dB better compared to DeblurGAN-v2.

RealBlur-R dataset: This paper compared and analyzed the results of DeepDeblur, DeblurGAN [5], DeblurGAN-v2 [11], SRN [41], DMPHN [44], MAXIM [20], MPRNet [45], and DeepRFT [46] on the RealBlur-R dataset.

Two training methods were used; RealBlur-R (GoPro) indicates training on the GoPro dataset and testing on RealBlur-R (to test generalization to real images); RealBlur-R indicates training and testing on RealBlur-R.

The average PSNR and average SSIM of different deblurring algorithms in the RealBlur-R dataset (GoPro) are shown in Table 3. The algorithm in this paper outperformed other defuzzification methods in terms of average PSNR and average SSIM. The algorithm in this paper achieved a PSNR of 36.08 dB, which was an improvement of 0.82 dB compared to DeblurGAN-v2.

The average PSNR and average SSIM of different deblurring algorithms on the RealBlur-R dataset are shown in Table 4. The algorithm in this paper outperformed other defuzzification methods in terms of average PSNR and average SSIM. The PSNR of the algorithm in this paper reached 39.65 dB, which was an improvement of 3.21 dB compared to DeblurGAN-v2.

The recovery results for the RealBlur-R dataset are shown in Figure 9. In the test results of the RealBlur-R dataset, two typical scenes were selected as subjective visual observations to compare the recovery effect of the algorithm in this paper. These two target image scenes were rich in content, and the images contained distinct edges and complex textures. Therefore, such complex scenes were a good test of the recovery ability of this method. As can be seen from the comparative results in the figure, the algorithm in this paper performed well in these samples and basically recovered the original details and colors of the image.

Real satellite images: The results of Restormer [47], DeblurGAN-v2 [11], DBGAN [40], and MAXIM [20] on real satellite remote sensing data were qualitatively analyzed.

The results of the real satellite image data recovery are shown in Figure 10. The images recovered by DBGAN and Restormer lacked many detailed textures, and DBGAN even generated false textures in some areas. The images recovered by DeblurGan-v2 and MAXIM had good visual effects but suffered from ghosting and unclear detail textures. In contrast, the method proposed in this paper produced visually pleasing images with fine and realistic textures for both vegetation and buildings.

Comparing the results of all algorithms on the RSDATA dataset, GOPRO dataset, RealBlur-R dataset, and real satellite blurred images, it can be seen that the algorithm in this paper had good performance metrics and visual results on all four datasets.

### 4.3. Ablation Study

#### 4.3.1. Effectiveness of the Feature Alignment Fusion Module

The feature alignment fusion module (FAFM) was removed from the generator structure, and the network was retrained on the RSDATA dataset using the same training scheme to verify the effectiveness of the FAFM.

As can be seen from Table 5, the average PSNR performance was improved from 33.13 to 37.20 and the average SSIM performance was improved from 0.81 to 0.91 after the addition of FAFM, verifying the effectiveness of FAFM in the deblurring process.

#### 4.3.2. Impact of the Feature Importance Selection Module on the Algorithm

The FISM of the feature alignment fusion module (FAFM) was removed from the generator network, and the network was retrained on the RSDATA dataset using the same training scheme to verify the effectiveness of the FISM.

In addition, this paper investigated the impact of feature selection on the generator network when different domains were used for FISM (channel domain (C_FISM), channel domain and one spatial domain (C_H_FISM, C_W_FISM), or channel domain and two spatial domains (C_H_W_FISM)).

As can be seen from Table 6, the average PSNR performance was improved from 35.18 to 37.20 and the average SSIM performance was improved from 0.82 to 0.91 with the addition of FISM, verifying the effectiveness of FISM in the deblurring process.

Table 7 shows the results of the quantitative evaluation of multidomain FISM. The SDD-GAN-recovered images using C_FISM scored the lowest in terms of the mean PSNR and mean SSIM metrics, indicating that the use of multidomain feature importance selection in FISM was effective in improving the quality of the recovered images. When C_H_W_FISM was used, the recovered images obtained high scores on the mean PSNR and mean SSIM metrics.

#### 4.3.3. Validity of the Color Feature Loss Function

Without using the color feature loss function, the network was retrained on the RSDATA dataset using the same training scheme to verify its effectiveness.
(17)$$L=w_{g}L_{g}+w_{p}L_{p}+w_{x}L_{x}$$

Table 8 shows the results of the quantitative evaluation. The SDD-GAN-recovered images using the color feature loss function scored higher than those without the color feature loss function in terms of both the mean PSNR and the mean SSIM metrics. The effectiveness of the color feature loss function was demonstrated.

## 5. Conclusions

For the remote sensing image deblurring task, this paper proposed a new end-to-end generative adversarial network (SDD-GAN) for single-image motion deblurring. The proposed feature alignment module (FAFM) and feature importance selection module (FISM) could successfully recover the texture and details of the blurred remote sensing images. The algorithm performed well on both self-constructed remote sensing datasets and publicly deblurred datasets, and the test comparison results on actual remotely sensed blurred images taken by satellite proved the effectiveness of the method proposed in this paper.

Next, we will conduct research on actual remote sensing blurred images, optimize the multiscale fusion architecture, improve the performance of the model in practical applications, and provide theoretical and experimental support for satellite remote sensing image blur recovery techniques.

## Figures and Tables

**Figure 1 sensors-22-07894-f001:**
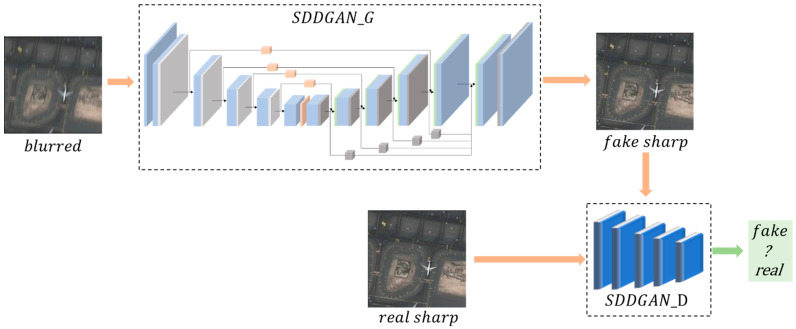
The SDD-GAN framework.

**Figure 2 sensors-22-07894-f002:**
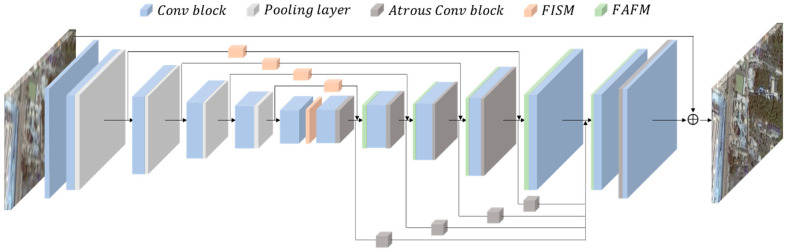
SDD-GAN generator structure diagram.

**Figure 3 sensors-22-07894-f003:**
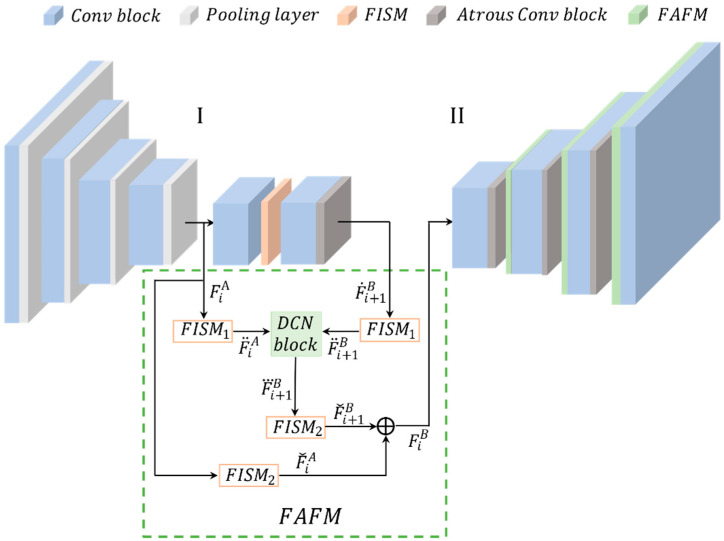
Diagram of the data flow of FAFM.

**Figure 4 sensors-22-07894-f004:**
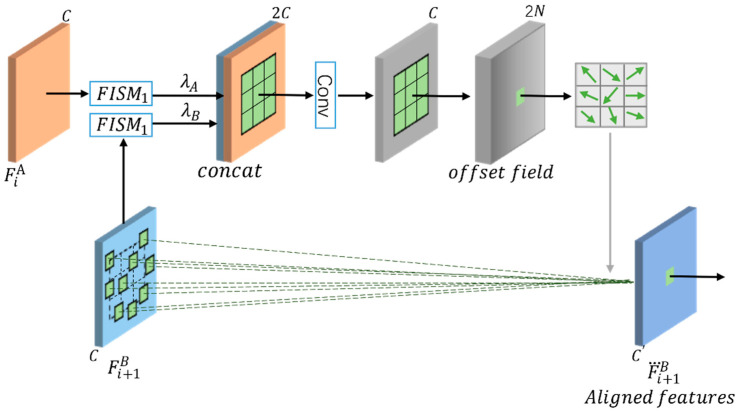
Schematic of offset calculation in FAFM.

**Figure 5 sensors-22-07894-f005:**
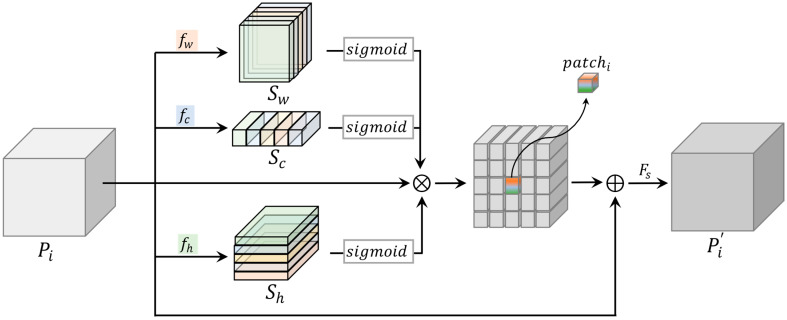
Diagram of the data flow of FISM.

**Figure 6 sensors-22-07894-f006:**
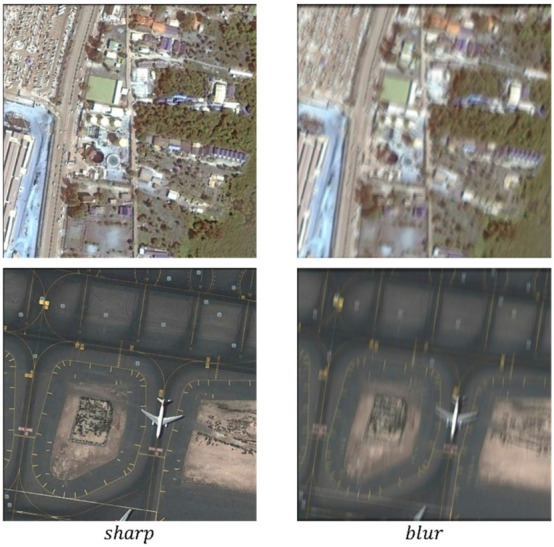
Example graph of RSDATA dataset.

**Figure 7 sensors-22-07894-f007:**
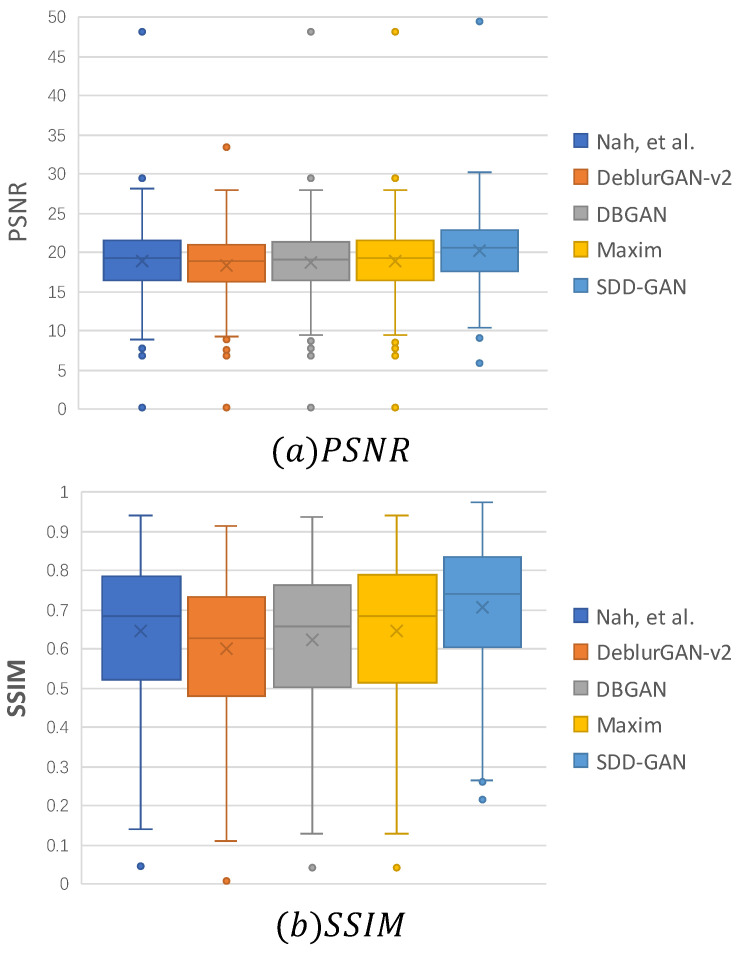
Distribution of the mean PSNR and SSIM for different deblurring algorithms on the RSDATA test set (trained on GoPro). The box plots show the maximum, upper quartile, median, lower quartile, and minimum values of the results from top to bottom.

**Figure 8 sensors-22-07894-f008:**
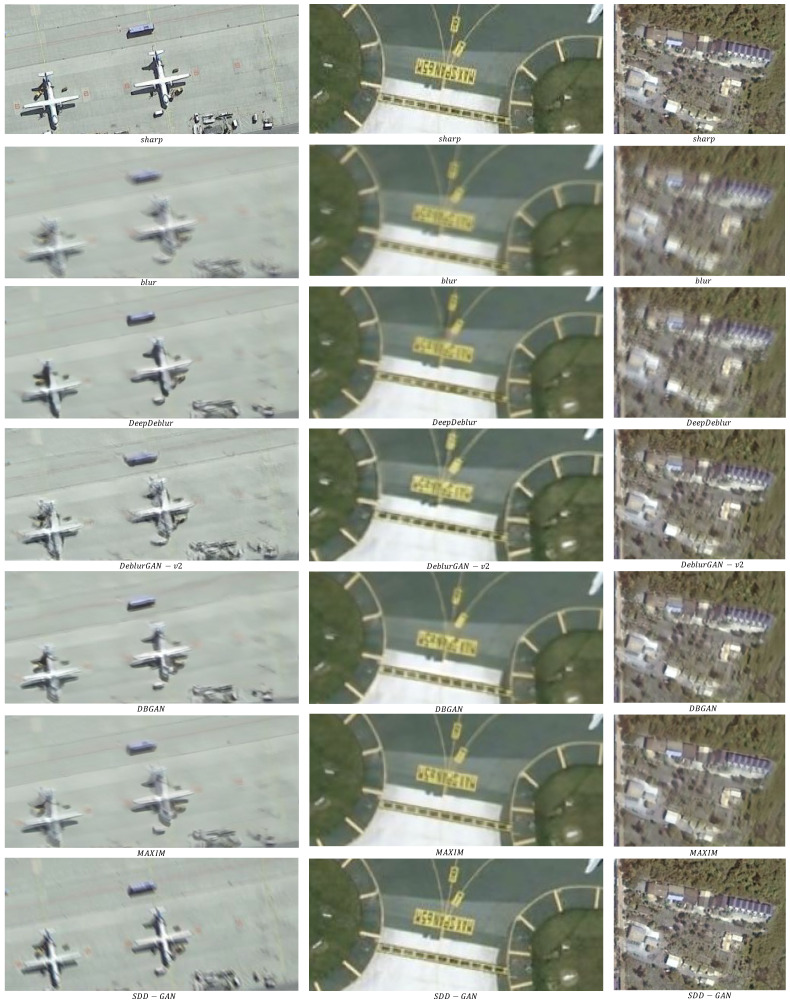
Results of different algorithms for image recovery in the RSDATA dataset.

**Figure 9 sensors-22-07894-f009:**
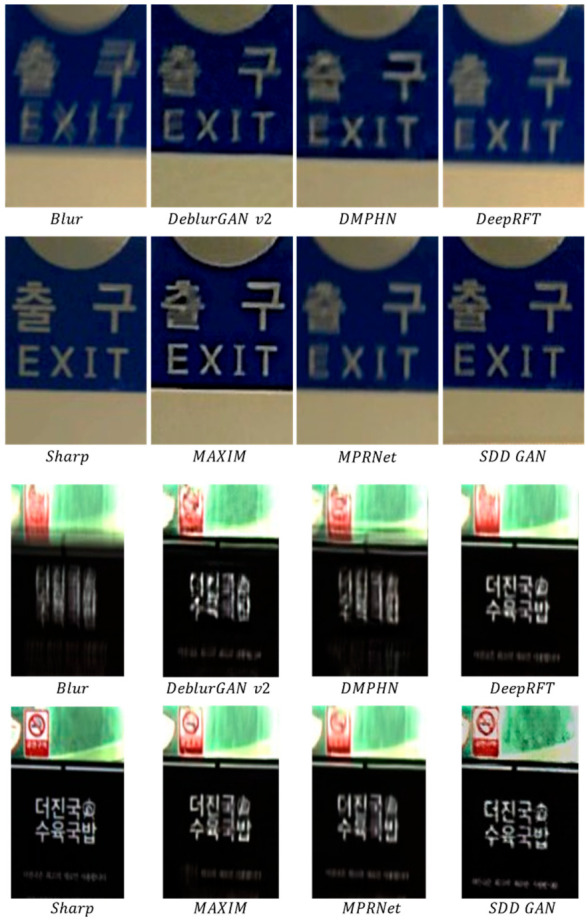
Comparison of recovery results for the RealBlur-R dataset.

**Figure 10 sensors-22-07894-f010:**
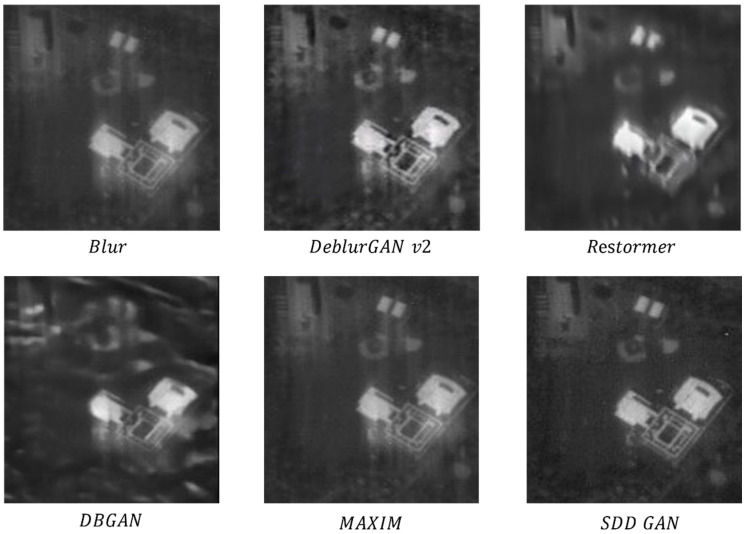
Test results for real satellite remote sensing blurred image.

**Table 1 sensors-22-07894-t001:** Average PSNR and SSIM for different deblurring methods on the RSDATA dataset (trained on the GoPro dataset). The highest scores are highlighted in red, and the second highest scores are highlighted in blue.

Train	Test	Method	PSNR	SSIM
GoPro	RSDATA	DeepDeblur [17]	18.85	0.64
GoPro	RSDATA	DeblurGAN-v2 [11]	18.36	0.60
GoPro	RSDATA	DBGAN [40]	18.67	0.62
GoPro	RSDATA	MAXIM [20]	18.86	0.64
GoPro	RSDATA	SDD-GAN	19.96	0.71

**Table 2 sensors-22-07894-t002:** Average PSNR and SSIM for different deblurring methods on the GoPro dataset. The highest scores are highlighted in red, and the second highest scores are highlighted in blue.

Method	PSNR	SSIM
DeblurGAN [5]	28.70	0.85
DeepDeblur [17]	29.08	0.91
SRN [41]	30.26	0.93
DeblurGAN-v2 [11]	29.55	0.93
STFAN [42]	28.59	0.86
BlurredItV [43]	30.58	0.94
SDD-GAN	30.95	0.94

**Table 3 sensors-22-07894-t003:** Average PSNR and SSIM for different deblurring methods on the RealBlur-R dataset (GoPro). The highest scores are highlighted in red, and the second highest scores are highlighted in blue.

Method	PSNR	SSIM
DeepDeblur [17]	32.51	0.84
DeblurGAN [5]	33.79	0.90
DeblurGAN-v2 [11]	35.26	0.94
SRN [41]	35.66	0.94
DMPHN [44]	35.70	0.94
MAXIM [20]	35.78	0.94
MPRNet [45]	35.99	0.95
DeepRFT [46]	36.06	0.95
SDD-GAN	36.21	0.96

**Table 4 sensors-22-07894-t004:** Mean PSNR and SSIM for different deblurring methods on the RealBlur-R dataset. The highest scores are highlighted in red, and the second highest scores are highlighted in blue.

Method	PSNR	SSIM
SRN [41]	38.65	0.96
DeblurGAN-v2 [11]	36.44	0.93
MAXIM [20]	39.45	0.96
SDD-GAN	39.65	0.97

**Table 5 sensors-22-07894-t005:** Mean PSNR and SSIM (with FAFM vs. without FAFM) on the RSDATA dataset. The highest scores are highlighted in red.

Method	PSNR	SSIM
SDD-GAN without FAFM	33.13	0.81
SDD-GAN	37.20	0.91

**Table 6 sensors-22-07894-t006:** Mean PSNR and SSIM (with FISM vs. without FISM) on the RSDATA dataset. The highest scores are highlighted in red.

Method	PSNR	SSIM
SDD-GAN without FIFM	35.18	0.82
SDD-GAN	37.20	0.91

**Table 7 sensors-22-07894-t007:** Average PSNR and SSIM on the RSDATA dataset (using C_FISM, C_W_FISM, C_H_FISM, or C_H_W_FISM). The highest scores are highlighted in red.

Method	PSNR	SSIM
SDD-GAN + C_FISM	35.51	0.87
SDD-GAN + C_W_FISM	35.69	0.90
SDD-GAN + C_H_FISM	35.74	0.90
SDD-GAN + C_H_W_FISM	37.20	0.91

**Table 8 sensors-22-07894-t008:** Average PSNR and SSIM (with L_C_ vs. without L_C_) on the RSDATA dataset. The highest scores are highlighted in red.

Method	PSNR	SSIM
SDD-GAN without Lc	37.20	0.91
SDD-GAN with Lc	39.88	0.92

## Data Availability

Not applicable.

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
