# Peer review of "Blind Deblurring of Remote-Sensing Single Images Based on Feature Alignment"

_sensors, 2022, doi:10.3390/s22207894_

Round 1
Reviewer 1 Report
The paper presents a new network architecture for image deblurring with unknown blur for remote sensing images. The article is very clear, it contains all details and shows that it outperforms currently art-of-state network algorithms for several data sets.
Minor comments are a clarification of how the camera shake is modeled.
In line 333, the function phi appears as a zero, and needs fixing.
Reviewer 2 Report
Dear Authors,
The paper presents interesting ideas, but I think this version is not publishable.
I focus on my comments following the paper presentation order.
1) line 63
Please, if you use the word "common", insert into the bibliography to some works that use this formalization.
2) line 239
What do you mean by "good semantic information"?
3) line 275
Please check the space between "F_i^B" and "Specially".
4) lines 290-292
Please describe "(,)" better.
5) line 311
Please describe "(.)" better.
6) Equation 9 (after line 236)
Please Please describe "~" better.
7) lines 364-365
You put a comma at the beginning of the new line. Please correct this minimal typo.
8) lines 377-380 plus equations 15 and 16
You introduce "Vobj" and "Lobj," but it is unclear how you use this information in your algorithm. Please pay particular attention to this description and try to explain the use of this information by providing well-documented theoretical and practical examples.
9) You refer to "Real Satellite Image" but it is not clear the dataset you use. The references to the other dataset are disseminated in the text.
I suggest you insert a short section or short paragraph with the references to all datasets you use.
The results you describe are good, and I would like to repeat the experiments. Can you please provide the source code or a compiled version? You could publish this code.
Best Regards
Round 2
Reviewer 2 Report
Dear Authors,
Regarding line 63, please help me understand where ref [5] presents your equation (1).
Regards
Round 3
Reviewer 2 Report
Dear Author,
The paper is accepted.
Regards